# Using a Simple Magnetic Adsorbent for the Preconcentration and Determination of Ga(III) and In(III) by Electrothermal Atomic Absorption Spectrometry

**DOI:** 10.3390/molecules28062549

**Published:** 2023-03-10

**Authors:** Yesica Vicente-Martínez, María José Muñoz-Sandoval, Manuel Hernández-Córdoba, Ignacio López-García

**Affiliations:** Department of Analytical Chemistry, Faculty of Chemistry, Regional Campus of International Excellence “Campus Mare Nostrum”, University of Murcia, 30100 Murcia, Spain

**Keywords:** gallium, indium, adsorption, magnetic dispersive solid-phase microextraction, electrothermal atomic absorption spectrometry

## Abstract

A solid-phase dispersive microextraction procedure has been developed using ferrite (Fe_3_O_4_), an inexpensive magnetic material, as an adsorbent for the separation and subsequent determination of Ga(III) and In(III). The ions were removed from an aqueous solution by adsorption on Fe_3_O_4_, which was next easily collected from the medium by the action of a magnet. The analytes were then desorbed using 50 µL of 2 M NaOH or 50 µL of a 4:1 mixture of 0.1 M EDTA and 2 M HNO_3_ for the determination of Ga(III) or In(III), respectively. The level of the elements in the desorption phase was measured by electrothermal atomic absorption spectrometry (ETAAS) by injecting 10 µL of this phase into the atomizer. The enrichment factor was 163, and detection limits of 0.02 and 0.01 µg L^−1^ were achieved for Ga(III) and In(III), respectively. The reliability of the procedure has been verified by means of standard reference materials and by means of standard additions. Results are given for waters, soils and samples obtained from various electronic devices. It is of note that the procedure could be the basis for a useful way of recovering these valuable elements from different matrices for reuse.

## 1. Introduction

Semiconductor manufacturing has become an important industry in some developing countries. Gallium and indium are elements widely used in the manufacture of integrated circuits, liquid crystal displays (LCDs), light emitting diodes (LEDs) for backlighting and lighting, and various semiconductors [1,2]. These metals are released into the environment during processes such as etching, wet polishing and cleaning or e-waste recovery operations, which can generate many potentially hazardous waste products [3,4]. Accidental industrial spills can result in high concentrations of Ga and In oxides and arsenides in water, which have acute and chronic toxic effects on aquatic organisms.

On the other hand, since demand for them is very high, the need to recover these elements from manufactured products at the end of their useful life is becoming more and more evident. In this sense, some processes have recently been proposed for the recovery of indium by pre-leaching in HNO_3_ + HF using a new adsorbent material consisting of mesoporous activated carbon [5] or an electrode with titanium dioxide and activated carbon [6]. In both cases, the recovery is close to 85%. The use of a pulsed laser has been proposed for the release of indium oxide from semi-conductors [7] to avoid lengthy multi-stage removal processes. In the case of gallium recovery, even the most recently proposed processes do not achieve a quantitative recovery. Thus, algae-based sorbents [8], biogenic elemental tellurium nanoparticles [9], synthetic zinc refinery residues [10] and H-clinoptilolite [11] have been successfully used as adsorbents. The reported removal efficiencies were, on average, 80% at low gallium concentrations (60% and 85% at 35 °C and 69 °C, respectively).

Whether or not recovery operations are carried out on these species, their presence in liquid effluents and wastes from production plants is increasing. Although these elements were initially thought to have little biological function and were not harmful to the human body, it has recently been reported that they can cause damage to human health [3]. In this regard, the Japanese Society for Occupational Health prescribed in 2007 that the maximum tolerable concentration of indium in serum should not exceed 3 μg L^−1^ [12]. In air, the German Ordinance on Hazardous Substances [13] sets a maximum permissible concentration of indium compounds at 0.1 µg m^−3^. As these elements accumulate in ecosystems, their toxic effects can be observed at the molecular, cellular and histological levels, even affecting the homeostasis of organisms [14]. Indium has no known biological function in living organisms [15]. The International Agency for Research on Cancer (IARC) classifies indium compounds in Group 2A, probably carcinogenic to humans [16]. Thus, indium contamination in the environment may pose a potential risk to public health via the food chain. On the other hand, the number of studies on the toxicity of Ga has been increasing until recently, and most of them support the conclusion that Ga has a very low toxicity [17], although this is highly dependent on the gallium compound and its way of entering the body [18].

The levels of these elements in soils vary considerably. The concentration of Ga in soils is usually between 3 and 70 mg kg^−1^. The total average content of In in unpolluted soils is between 0.01 and 0.5 mg kg^−1^. The distribution of In and Ga in soils mainly indicates their association with the clay fraction. In water samples, Ga and In concentrations are very low. The average concentration of Ga in seawater is 30 ng L^−1^; in rivers, it ranges globally from 1 to 22 ng L^−1^. In the oceans, the concentration of In has been estimated to be about 0.1 ng L^−1^ [19]. In most environmental samples, the concentration of these elements is close to the detection limits of analytical techniques, hence the interest in having sensitive and selective methods for their determination.

Ga(III) can be determined by molecular absorption spectrometry using ligands that form compounds with high molar absorptivity [20,21,22,23], whose sensitivity can be improved by coupling to a liquid- [24] or solid-phase extraction procedure [25]. If the compound resulting from the interaction of Ga(III) with the ligand emits fluorescence, sensitivity is further enhanced [26]. When combined with solid-phase extraction, a limit of detection (LOD) of 3.1 µg L^−1^ can be achieved with Vis-UV detection [25].

However, atomic methods have undoubtedly been the most widely used for the determination of Ga(III). In the case of atomic absorption, flame atomization (FAAS) [27,28] and electrothermal [29,30,31] techniques (ETAAS) have been proposed, although the narrow linearity range characteristic of this technique has led many authors to use atomic emission spectrometry with optical detection (ICP-OES) [32,33,34,35] or mass spectrometry (ICP-MS) [36,37] for the purpose. These latter techniques are expensive, mainly due to the high argon consumption. This fact, together with interferences caused by overlapping spectral lines in ICP-OES or the high salt content of some samples, which is incompatible with ICP-MS, has led several authors to propose electrochemical methods, mainly based on anodic stripping voltammetry, that achieve similar detection limits [38,39,40,41,42].

Most of the recently proposed methods using atomic detection techniques couple preconcentration processes in different media. For example, supercritical fluid extraction [31], silica nanoparticles packed in a microcolumn [43], solid-phase extraction with a cation exchanger [34], non-ionic [25,44] or chelating [45], modified resins [46], halloysite nanotubes [30], silica gel modified with gallic acid [47] or simply extraction with organic solvents [28], at cloud point [27] and in ionic liquids [48].

In the case of In(III), analytical methods for trace determination include ETAAS [30,45,49,50,51,52,53,54], FAAS [27,55,56], ICP-OES [57,58,59,60] and ICP-MS [61,62,63]. To a lesser extent, because they do not achieve sufficient sensitivity, Vis-UV molecular absorption spectrometry [46,47,64,65,66,67,68] and electrochemical methods are used [40,41]. Preconcentration techniques are used whenever low detection limits are required to avoid interferences. Few microextraction techniques have been proposed for the determination of indium. Most of them use liquid–liquid dispersive microextraction [65,67], floating drop extraction [69], ultrasound-assisted extraction [70] or cloud point extraction [27,71]. Solid-phase extraction, in its different variants, has received more attention. Silica gel [49], chelating resins with suitable ligands [45,46,52,54,59], coprecipitation with chitosan [53] and sodium dodecyl sulphate modified activated carbon [50] have been proposed.

In recent years, dispersive micro-solid-phase extraction (DMSPE) has gained prominence due to its sensitivity and speed. In DMSPE, a small amount of sorbent is dispersed in the solution to induce an immediate interaction between the metal ions and the nanomaterial. In this sense, the use of halloysite nanotubes [30] for Ga(III) and In(III) separation and graphitic carbon nitride [60], oleic acid coated magnetic particles [56] and bentonite clay [72] for In(III) determination have been proposed.

In this work, the interaction of indium and gallium with ferrite, an easily synthesized and cheap magnetic material, has been studied. Although several processes using solid particles as adsorbent material in microextraction processes have been proposed, none is as easy to synthesize as the one proposed in this work, with the added advantage of easy separation of the adsorbent material using a magnetic field. After the adsorption on ferrite, the ions were desorbed in a different medium and submitted to ETAAS. The results obtained with this study show that, on the one hand, it was possible to quantify both species at trace concentrations, achieving high enrichment factors and low LODs for both species. On the other hand, the study of the adsorption of the species on ferrite is a simple and economical proposal for the recovery of these valuable species.

## 2. Results and Discussion

### 2.1. Effect of pH

The effect of pH on the retention of Ga(III) and In(III) ions by ferrite was studied in the range 3–10 by using different buffer solutions, and the results are shown in Figure 1. As can be seen from curve a, corresponding to Ga(III), the retention was practically total over the whole range studied, whereas the retention of In(III) reached a maximum in the range 7.5 < pH < 8.5. In order to standardize the experimental procedure, the use of pH = 8 is recommended. The surface charge of the ferrite starts to become positive below pH = 6.5 (Appendix A). Therefore, at the pH found to be optimal for the retention of In(III) and Ga(III), the surface charge of the adsorbent material is negative.

In the case of gallium, the existence of polynuclear species as opposed to mononuclear species has been described. Indeed, gallium(III) ions give rise to monomeric Ga(OH)_3_ and Ga(OH)_4_^−^ hydroxides together with polynuclear hydroxides such as Ga_3_(OH)_11_^2−^, Ga_4_(OH)_11_^+^ and Ga_6_(OH)_15_^3+^ [73,74]. The tendency of Ga(III) to form polyionic hydroxides and the difficulty in determining mononuclear species seems to have been demonstrated. Indeed, the presence of Ga_6_(OH)_15_^3+^ formed around pH = 2.8 has been described. Appendix A shows the gallium distribution taking into account these hydrolysis phenomena. These polynuclear hydroxides, together with Ga(OH)_3_, have been termed colloidal solid gallium [73,75,76]. Since the predominant species in the pH range studied is Ga(OH)_3_, we assume that this must be the species retained on the surface, although we must also consider the possible adsorption of gallium in the form of positively charged polynuclear hydroxylated complexes. Although as the degree of polymerization increases, the polyhydroxylated gallium complexes reduce their positive charge by converting the OH^−^ bridges into O^2−^ bridges, their ability to interact with anionic surfaces is clear and this would justify their adsorption on ferrite at pH values above 7.

In the case of In(III), the distribution of species at different pH values (Appendix A) indicates that the predominant species in the pH range studied is In(OH)_3_. However, the presence of chlorohydroxo complexes, mainly In(OH)Cl^+^ and to a lesser extent In_2_(OH)Cl_4_^+^, cannot be excluded. It is therefore assumed that it is the neutral In(OH)_3_ species which adsorb on the ferrite surface.

### 2.2. Effect of the Adsorbent Amount and the Time of Contact

The amount of ferrite required to achieve a quantitative retention of the analytes was studied. For this study, concentrations as high as 200 µg L^−1^ were used since they ensure that the results obtained will also be adequate for solutions with much lower concentrations and, at the same time, guarantee that even if some active sites are occupied by other species, there will still be enough available for the analytes.

The amount of adsorbent material was varied between 0 and 5 µg, corresponding to suspension volumes of up to 300 µL. Figure 2A shows the results obtained for the retention of Ga(III), those of In(III) being very similar. As shown in the graph, the retention becomes maximum and constant from about 1 mg of adsorbent. We conclude that the presence of only 1.7 mg of adsorbent material is sufficient to retain the tested concentrations of these ions. It can be predicted that at much lower analyte concentrations, as would be the case if the proposed preconcentration procedure is applied, the number of active sites for adsorption is guaranteed. It is therefore recommended that 100 µL of the adsorbent suspension is used for a 10 mL sample containing the analytes of interest.

On the other hand, the contact time between the adsorbent and the solution containing the analytes was also studied. Figure 2B shows the influence of the contact time between these solutions and the adsorbent material when working at pH = 8 and room temperature. The time was counted from the moment the ferrite was added until the neodymium magnet was applied to separate the solid phase. As can be seen, the retention increased rapidly during the first 2 min and it was complete in less than 4 min. This rapid increase in adsorption efficiency can be attributed to the large amount of vacant active sites on the adsorbent surface. When working with much lower concentrations of these ions to achieve a preconcentration effect, the time required will be much shorter, as already described for similar processes [77,78]. The results obtained for the retention of In(III) under the recommended experimental conditions were similar to those obtained for Ga(III). Therefore, a contact time of 5 min was adopted as sufficient to ensure quantitative retention of both species. Up to 25 mL of sample can be treated with 200 µL of ferrite suspension and complete adsorption can be achieved in less than 10 min without changing the volume of desorption medium.

### 2.3. Study of the Desorption Conditions

The desorption of the analytes in a microvolume of liquid phase compatible with the detector is one of the main objectives to be achieved in a microextraction process. In this respect, the behavior of Ga(III) was very different from that of In(III). Figure 3 shows the desorption rates obtained for Ga(III) in different media. In all cases, 100 µL was used as the desorption volume. As can be seen, all media were basic, as desorption in acidic media was very low. Each medium was tested with and without the use of ultrasound and at two temperatures: room temperature and 80 °C. Desorption was tested in: (a) 0.2 M NaOH; (b) 0.2 M NH_4_OH; (c) 0.1 M tetramethylammonium hydroxide and (d) 0.1 M ethylenediaminetetraacetic acid disodium salt (EDTA) at pH = 12. The best results were obtained using 0.2 M sodium hydroxide and heating at 80 °C for 5 min in an ultrasonic bath. Under these conditions the desorption was quantitative.

For In(III), different media were tested at two temperatures (room temperature and 80 °C) and in the presence or absence of ultrasound. The media used were: (a) 0.2 M sodium hydroxide; (b) 0.005% *m*/*v* dithizone; (c) 0.05% *m*/*v* dithizone; (d) 0.01 M phenanthroline; (e) mixture of HF and HNO_3_ acids at 8% in HF and 4% in HNO_3_; (f) 0.1 M EDTA and (g) a medium consisting of 80 µL of 0.1 M EDTA and 20 µL of 2 M HNO_3_. As shown in Figure 4, the best results were obtained using EDTA in acidic medium. Quantitative desorption was achieved by using 80 µL of 0.1 M EDTA and 20 µL of 2 M HNO_3_ as desorption media and applying ultrasound for 2 min.

On the other hand, different volumes of desorption phase were assayed. As could be expected, as the desorption volume increased, and the analyte concentration, and thus the analytical signal, decreased. Finally, looking for a compromise between maximum (total) desorption and minimum dilution of the analytes, 50 µL was selected as the volume to be used for desorption.

### 2.4. Optimization of the ETAAS Parameters

The resonance line suggested in the software of the spectrometer for the determination of gallium is 287.424 nm. However, this wavelength is not suitable for the purpose since iron presents an absorption line at 284.417 nm. Despite this line having a low sensitivity, it affects the analytical signal (see Figure 5A) due to the relatively large amount of iron involved. To avoid the deleterious effect, a secondary line of gallium, namely 294.346 nm, is recommended here. Using this line, no interference effect was observed and the selectivity was excellent (Figure 5B).

In the case of indium determination, the resonance wavelength of 303.935 nm was chosen since the signal obtained at this wavelength is not affected by the presence of iron (Figure 6).

The convenience of using a chemical modifier for the atomization stage was also considered [79]. The results showed that for the determination of gallium, the use of palladium nitrate as a chemical modifier was suitable to improve sensitivity and to obtain signals with low background absorption.

Similar studies were carried out for the indium determination, testing different common chemical modifiers, and again palladium proved suitable. Therefore, the procedure includes the injection into the atomizer of 10 µL of a 500 mg L^−1^ Pd(II) solution as a chemical modifier.

The calcination and atomization temperatures were optimized in the usual way, looking for low background and well-defined atomization profiles, and the results are included in Table 1 where the rest of parameters are summarized.

### 2.5. Analytical Figures of Merit: Applications

Using the conditions optimized, a linear behavior was found between the peak area signal and the concentration of gallium or indium species in solution in the range 0.1–1.5 and 0.05–1 µg L^−1^, respectively. The slopes of the calibration lines were 0.414 ± 0.005 and 0.683 ± 0.007 s µg^−1^ L, and the limits of detection, calculated from the threefold standard error of the regression, were 0.02 and 0.01 µg L^−1^ for gallium and indium, respectively. The relative standard deviation of the measurements within the calibration range was between 4.2 and 4.8%.

The enrichment factor was calculated from the ratio between the slopes measured after application of the microextraction procedure and those obtained directly, and was found to be 163 ± 2 when the volume of aqueous phase was 10 mL and a volume of 50 µL of the corresponding solvent was used for desorption.

Table 2 shows the main procedures reported in the recent literature for the determination of both analytes and involving liquid–liquid or liquid–solid microextraction processes for preconcentration. As can be seen, the procedure studied in this work successfully competes with those already reported since it allows a high preconcentration factor to be achieved with minimal sample consumption, furthermore involving an affordable and selective technique for quantification.

To test its usefulness and reliability, the procedure studied was applied to the determination of gallium and indium ions in four water samples (tap water, sea water and two bottled water samples). The reliability was first assessed by applying the standard additions method and, as shown in Table 3, the recoveries of the spiked samples were satisfactory at the 95% confidence level.

The reliability was further confirmed by measuring the gallium and indium content in several standard reference materials (SRMs): NIST SRM 2711, NCS DC 73319a, TM-25.4 and TMDA-62.2. As two of them did not contain indium, they were analyzed before and after their fortification with indium (0.5 µg L^−1^) and relative recoveries of 93 to 107% and RSDs of 7 to 9% were obtained. The analytical data obtained are summarized in Table 4. These results demonstrate that the proposed method, together with the instrumentation used, is suitable for the analysis of soil, sediment and water samples.

The procedure studied was also applied to the determination of gallium and indium in various electronic devices, namely three mobile phones of very different dates of manufacture, the screens of two non-touch laptop computers and the diodes of several LED light bulbs. All these devices were treated according to the procedure described in Section 3.3. The levels of the two elements found are shown in Table 5. For LCD panels, the values shown correspond to the contents in the panels after removal of the polymer frames protecting the LCD structure (which were not attached to the LCD panel). The polymeric polarizing film, which remained strongly adhered to the glass surface, was then removed in an acetone bath. This made it easier to grind the material.

## 3. Materials and Methods

### 3.1. Materials and Instrumentation

All the solutions were prepared with deionized water (resistivity > 18 MΩ cm) purified using a Millipore system (Millipore, Bedford, MA, USA). Glassware was washed with 10% *v*/*v* nitric acid solution and rinsed with this pure water before use.

Ga(III) and In(III) standards were prepared from 1 g L^−1^ standard solutions supplied by Panreac (Barcelona, Spain). FeCl_3_·6H_2_O, FeCl_2_·4H_2_O (Sigma, St. Louis, MO, USA) and ammonium hydroxide (Merck, Darmstadt, Germany) were used for the synthesis of the ferrite particles. A 10 g L^−1^ solution of Pd(NO_3_)_2_ (Sigma-Aldrich) was used as a chemical modifier. The rest of the reagents were from Merck or Sigma-Aldrich.

All atomic absorption measurements were carried out with a high-resolution continuous-source atomic absorption spectrometer (HR-CS-AAS, model contrAA 700) provided by Analytik Jena AG (Jena, Germany) and equipped with a transversely heated graphite atomizer. The furnace heating program and the optimized experimental conditions are summarized in Table 1.

A 50 W ultrasonic bath model LT-80pro (Terratech, Barcelona, Spain) with temperature control was used for thermal and ultrasonic treatments. Vortexing was carried out with a Reax Top from Heidolph (Schwabach, Germany). For centrifugation, a Hettich Eba 200 centrifuge (Tuttlingen, Germany) with rotor and adapter for 15 mL conical bottom tubes at up to 6000 rpm was used. Samples, when necessary, were ground in an agate ball mill model MM2 from Retsch (Haan, Germany). Mineralization of the samples was carried out using an Anton Paar Multiwave 3000 microwave oven (Graz, Austria). For calcination of the samples, when required, a Hobersal GB 7-122 muffle furnace (Barcelona, Spain) was used.

Magnetic discs (Supermagnete, Gottmadingen, Germany) were used for magnetic separation during the synthesis of the nanoparticles and their removal from the aqueous medium.

### 3.2. Preparation of the Magnetic Material

Fe_3_O_4_ particles were prepared by the coprecipitation procedure described in [83], with slight modifications. In this process, 20 mL water (from which the oxygen has been removed by a stream of nitrogen) was heated to 80 °C and then 0.56 g FeCl_3_·6H_2_O and 0.2 g FeCl_2_·4H_2_O were added while stirring continuously. After dissolution of the solids, 2 mL of concentrated ammonium hydroxide was added. The black precipitate that appeared was shaken at 80 °C for 1 h. The magnet was then applied so that the ferrite particles were attracted to the wall of the tube and the supernatant was removed by decantation. The Fe_3_O_4_ particles were then washed with 20 mL portions of water and decanted in the same way until the pH of the solution was neutral. Finally, the iron particles (about 0.34 g Fe_3_O_4_) were suspended in 20 mL of water and kept refrigerated for use.

Particle size and zeta potential data for the adsorbent materials were obtained, and the results are given as Appendix A, respectively). Appendix A shows the Field Effect Scanning Electron Microscopy (FESEM) image corresponding to the surface of Fe_3_O_4_. In addition, Appendix A shows the associated Energy Dispersive X-ray (EDX) spectra and the atomic concentration table for Fe_3_O_4_. The former displays strong peak signals for Fe and O. Signals for Al, Cu, Si and C correspond to the sample holder and the grid employed to obtain the FESEM images.

### 3.3. Samples Treatment

Water samples were filtered through a 0.45 µm syringe filter and then stored in polyethylene bottles at 4 °C for a maximum of 7 days.

Soil samples were treated by microwave-assisted wet digestion. Approximately 500 mg of the powdered certified reference material was placed in the PTFE container of the microwave system and 1 mL of 30% H_2_O_2_ was added to wet the sample. Then, 4 mL of 65% HNO_3_ and 2 mL of 40% HF were added. After digestion (180 °C for 20 min), the clear solutions were transferred to calibrated 10 mL flasks and diluted according to the concentration of the analytes.

Light emitting diodes and mobile phone and laptop screens were first manually disassembled and separated into the different fractions. The polarizing polymer film, which remained strongly adhered to the surface of the display glass, was removed using an acetone bath. The samples were then cut into small pieces (approximately 1 × 1 cm) and ground in a ball mill for 120 min. Finally, the LED powder obtained was sieved through a 100 µm sieve [84]. To determine the total Ga and In contents, the powder was mixed with Na_2_CO_3_ in a 1:1 ratio and calcined (800–1100 °C), the furnace temperature being increased linearly at 10 °C min^−1^ [85]. The ashes were dissolved with 0.5 M H_2_SO_4_ and appropriately diluted depending on the amounts of indium and gallium involved.

For the leachates, the soil samples were treated with 1 M H_2_SO_4_ using a liquid to solid ratio of 50:1 at 90 °C and mechanical stirring for 60 min [85].

### 3.4. Procedure for the Determination of Gallium and Indium

The sample or standard solution was adjusted to pH = 8 with HNaCO_3_ and the addition of small volumes of 0.1 M HNO_3_ or NaOH, if necessary. After 5 min manual shaking, 100 µL of the Fe_3_O_4_ suspension prepared as indicated above was added. The adsorbent phase was separated with the neodymium magnet and the supernatant was discarded. For the determination of gallium, 50 µL of 2 M NaOH was added to the solid phase and the mixture was vortexed for 30 s. It was then placed in an ultrasonic bath at 80 °C for 5 min, the magnet being applied again to separate the solid. Next, 10 µL aliquots of the supernatant phase were taken and injected according to the heating program given in Table 1.

For the determination of indium, desorption from the adsorbent was carried out using a medium consisting of 80 µL of 0.1 M EDTA and 20 µL of 2 M HNO_3_. After vortexing for 30 s and holding the mixture at 80 °C for 5 min, the magnet was reapplied and the supernatant (10 µL) injected into the atomizer.

## 4. Conclusions

Electrothermal atomic absorption spectrometry in conjunction with a simple magnetic dispersive solid-phase microseparation step has proved suitable for the purpose of determining low concentrations of gallium and indium. The adsorbent used (Fe_3_O_4_) is cheap and easy to obtain and, due to its magnetic properties, can be conveniently removed from the medium by the action of a magnet, thus avoiding the need for a centrifugation step. It is interesting to note that, in addition to the sensitive and selective analytical procedures here studied, the characteristics of this simple magnetic material open the door to its use for the recuperation or removal of the gallium and indium present in some residues.

## Figures and Tables

**Figure 1 molecules-28-02549-f001:**
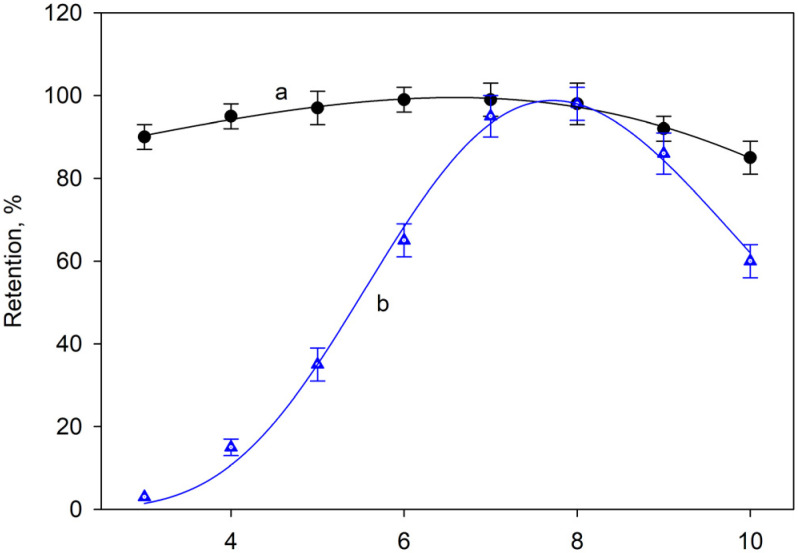
Effect of pH on Ga(III) and In(III) retention (curves a and b, respectively). Error bars correspond to the standard deviation of three determinations.

**Figure 2 molecules-28-02549-f002:**
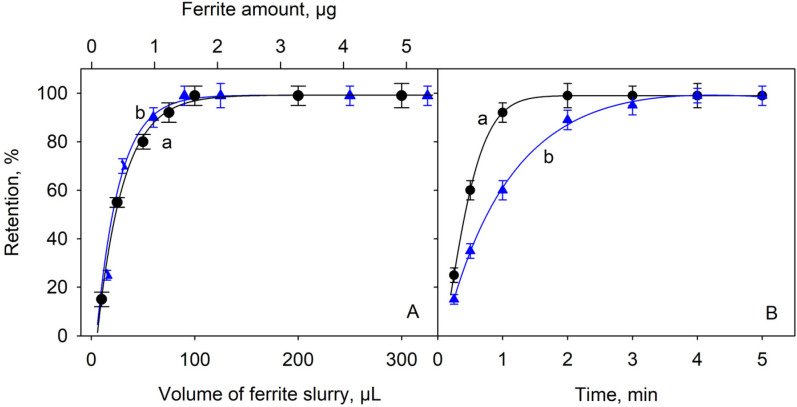
(**A**) Effect of contact time between Ga(III) and In(III) solutions (curves a and b, respectively) and adsorbent material on retention efficiency; (**B**) effect of the amount of adsorbent on the retention of Ga(III) and In(III) (curves a and b, respectively).

**Figure 3 molecules-28-02549-f003:**
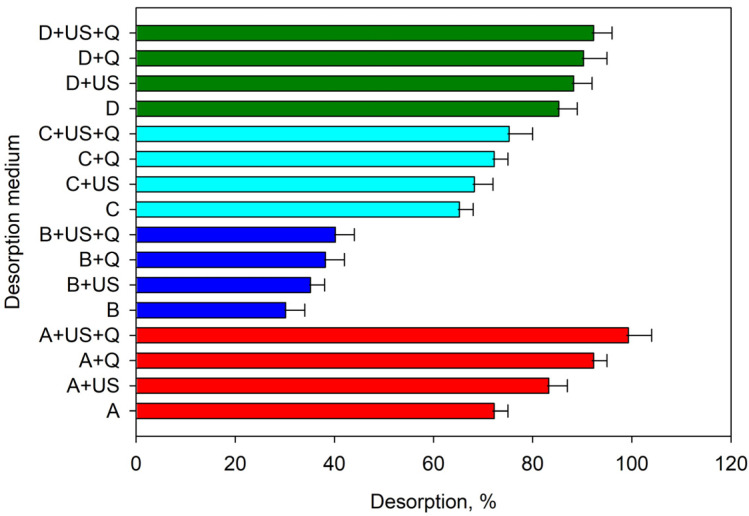
Effect of the desorption medium on the releasing of Ga(III) from the adsorbent to the aqueous phase. Media A (red), B (dark blue), C (blue) and D (green) correspond to 0.2 M NaOH, 0.2 M NH4OH, 0.1 M TMHA and 0.1 M EDTA solutions, respectively. The terms US and Q correspond to the use of an ultrasonic bath for 5 min and heating at 80 °C, respectively.

**Figure 4 molecules-28-02549-f004:**
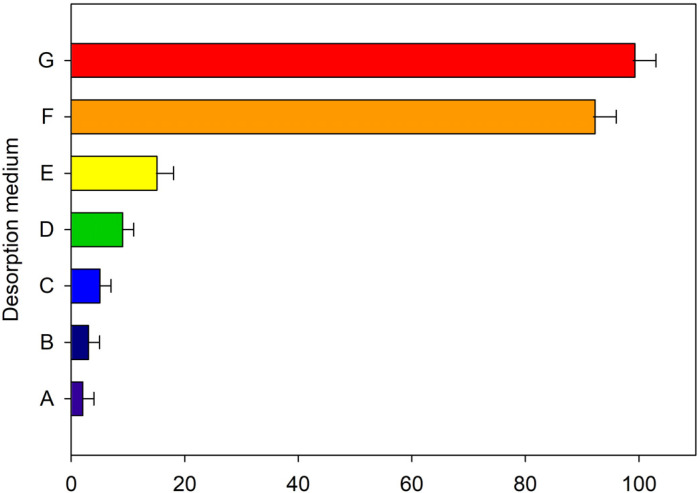
Effect of the desorption medium on the releasing of In(III) from the adsorbent to the aqueous phase. Media A–G correspond to 0.2 M NaOH, 0.005% *m*/*v* dithizone, 0.05% *m*/*v* dithizone, 0.01 M phenanthroline, HF + HNO_3_; 0.1 M EDTA and a medium consisting of 80 µL of 0.1 M EDTA and 20 µL of 2 M HNO_3_.

**Figure 5 molecules-28-02549-f005:**
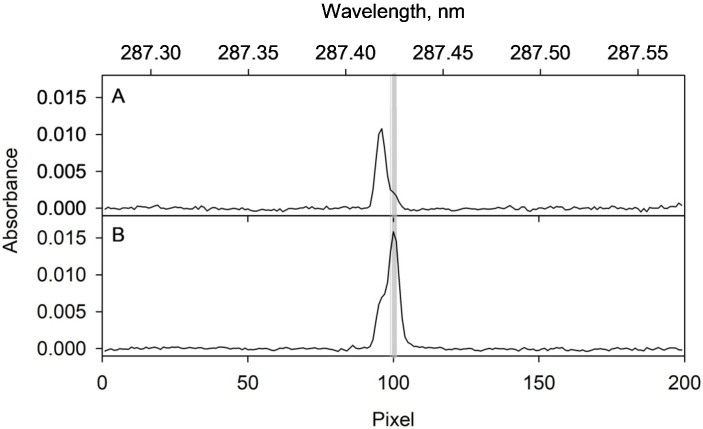
Average absorbance measured during the integration time at each observation pixel for the Ga wavelength at 287.424 nm. (**A**) corresponds to the signal from the desorption supernatant of a blank assay. (**B**) corresponds to the signal obtained for the desorption supernatant resulting from the application of the proposed procedure to a 2 µg L^−1^ Ga(III) solution. As can be seen, the absorption bands of Fe(III) and Ga(III) overlap. The grey band corresponds to pixels 99, 100 and 101, from which the analytical signal is calculated.

**Figure 6 molecules-28-02549-f006:**
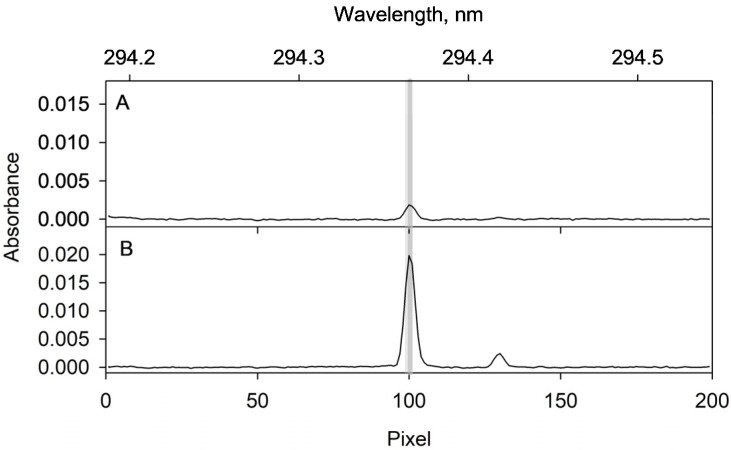
Absorbance averaged over the integration time at each observation pixel for the Ga measurement at 294.364 nm. (**A**) corresponds to the signal from the desorption supernatant of a blank assay. (**B**) corresponds to the signal obtained for the desorption supernatant when the proposed procedure is applied to a 2 µg L^−1^ Ga(III) solution. The grey band corresponds to pixels 99, 100 and 101 from which the analytical signal is calculated. The small absorption band due to gallium at 294.417 nm is also visible.

**Table 1 molecules-28-02549-t001:** Instrumental parameters and heating program for Ga and In.

Parameter			
Wavelength, nm		294.364 (Ga)/303.935 (In)	
Slit, nm		0.7	
Atomizer		Transversal with L’Vov platform	
Background correction		Zeeman effect	
Injected sample volume, µL		10	
Chemical modifier		10 µL of 500 mg L^−1^ Pd(II) solution	
Sample volume, mL		10	
**Heating program**			
Step	Temperature, °C	Ramp, °C s^−1^	Hold, s
1: Dry	110	10	20
2: Dry	130	9	10
3: Calcination	1300	300	15
4: Atomization ^a,b^	2300 (Ga)/2200 (In)	2300 (Ga)/1500 (In)	5
5: Clean	2450	500	4

^a^ Internal argon flow stopped 5 s before. ^b^ Reading step.

**Table 2 molecules-28-02549-t002:** Characteristics of some procedures reported for the determination of Ga and In with the use of microextraction techniques.

Analyte	Separation Technique	TD	V	FE	LOD	Ref.
Ga, In	SPE with aminated silica gel functionalized with HBAAS	FAAS	1000	200	4.1; 1.55	[80]
Ga, In	SPE with aminated silica gel functionalized with gallic acid	FAAS	50	250	5.8; 1.8	[47]
In	SPE with TiO_2_	Vis-UV	25	25	450	[68]
In	SPE with Chromosorb 108 and bathocuproine disulphonic acid	ETAAS	100	30	0.012	[54]
In	SPE with SDS-modified activated carbon	ETAAS	200	363	0.0002	[50]
In	DMSPE with exfoliated graphitic carbon nitride nanosheets	ICP-OES	10	75	0.32	[60]
Ga, In	USA-DMSPE with halloysite nanotubes	ETAAS	10	33; 37	0.02; 0.01	[30]
In	VASPME with oleic acid coated ferrite	FAAS	30	44	6.02	[56]
In	DLSME with chitosan	ETAAS	500	100	0.4	[53]
Ga	CPE with 8-hydroxyquinoline and Triton X-114	ICP-MS	50	20	1.2	[71]
Ga, In	CPE with gallic acid on Triton X-114	FAAS	25	54; 48	3.5; 1.25	[27]
Ga, In	CPE with 5-Br-PADAP in the presence of Triton X-100	ICP-OES	25	12	0.7; 0.3	[81]
Ga	DLLME with APDC	XRF	5	250	1.7	[82]
In	DLLME with PAN	Vis-UV	10	160	0.3	[65]
Ga	USA-DLLME with PAN in chlorobenzene	FAAS	40	124	70	[70]
In	SFOD-DLLME with dithizone and 1-undecanol	Vis-UV	10	36	9	[67]
In	SFOD-DLLME with PAN and 1-undecanol	ETAAS	25	62,5	0.05	[69]
Ga, In	MDSPE with ferrite	ETAAS	10	163	0.02; 0.01	TW

TD: detection technique; V: sample volume, mL; FE: enrichment factor; LOD: limit of detection, µg L^−1^; Ref: reference; TW: this work; SPE: solid-phase extraction; HBAAS: 2-hydroxy-5-(2-hydroxybenzylideneamino)benzoic acid; CPE: cloud point extraction; 5-Br-PADAP: 2-(5-bromo-2-pyridinazo)-5-diethylaminophenol; SDS: sodium dodecyl sulphate; DMSPE: dispersive micro-solid-phase extraction; USA-DMSPE: ultrasonic-assisted dispersive micro-solid-phase extraction; VASPME: vortex-assisted solid-phase microextraction; DLSME: dispersive liquid–solid microextraction; XRF: X-ray fluorescence; PAN: 1-(2-pyridinazo)-2-naphthol; USA-DLLME: ultrasonic-assisted liquid–liquid dispersive microextraction; SFOD-DLLME: floating solid droplet DLLME; MDSPME: magnetic dispersive solid-phase microextraction.

**Table 3 molecules-28-02549-t003:** Results of Ga(III) and In(III) in real water samples (RWS).

	Ga(III), µg L^−1^			In(III), µg L^−1^		
RWS	Added	Found ^a^	Rec., %	Added	Found ^a^	Rec., %
M1	0 0.2	<LOD 0.19 ± 0.02	- 95 ± 6	0 0.1	<LOD 0.10 ± 0.02	- 103 ± 6
M2	0.4 0	0.42 ± 0.02 <LOD	105 ± 7 -	0.3 0	0.29 ± 0.02 <LOD	98 ± 7 -
	0.2 0.4	0.21 ± 0.03 0.38 ± 0.02	105 ± 6 95 ± 5	0.1 0.3	0.09 ± 0.02 0.31 ± 0.03	92 ± 5 104 ± 5
M3	0 0.2	<LOD 0.21 ± 0.02	- 105 ± 5	0 0.1	<LOD 0.11 ± 0.02	- 110 ± 6
M4	0.4 0	0.39 ±0.03 <LOD	98 ± 6 -	0.3 0	0.32 ± 0.03 <LOD	107 ± 7 -
	0.2 0.4	0.18 ± 0.03 0.41 ± 0.03	90 ± 6 103 ± 4	0.1 0.3	0.09 ± 0.02 0.31 ± 0.04	93 ± 6 103 ± 5

M1: drinking water from the supply network; M2: sea water; M3 and M4: bottled water. ^a^ mean value ± standard deviation of three determinations.

**Table 4 molecules-28-02549-t004:** Results of Ga(III) and In(III) in reference materials.

	Ga(III), µg g^−1^		In(III), µg g^−1^	
SRM	Certificate	Found ^a^	Certificate	Found
NIST SRM 2711 ^b^ (soil)	15	15.8 ± 0.4	1.1	1.2 ± 0.1
NCS DC 73319a (soil)	18 ± 1.4	17.9 ± 0.3	0.12 ± 0.02	0.12 ± 0.01
	**Ga(III), µg L^−1^**		**In(III), µg L^−1^**	
**SRM**	**Certificate**	**Found**	**Certificate**	**Found**
TM-25.4 (water)	32.6 ± 2.8	33.2 ± 0.5	-	<LOD
TMDA-62.2 (water)	9.03 ± 0.73	8.98 ± 0.23	-	<LOD

^a^ mean value ± standard deviation of three determinations. ^b^ noncertified values.

**Table 5 molecules-28-02549-t005:** Determination of gallium and indium in optoelectronic devices.

Sample Treatment	Ga(III), µg g^−1^			In(III), µg g^−1^		
Calcination	Added	Found ^a^	Rec., %	Added	Found ^a^	Rec., %
LED light bulb	- 20	730 ± 3 749 ± 4	- 99.8 ± 3.7	- 20	20 ± 2 41 ± 2	- 102.5 ± 4.7
LCD Phone 1	- 20	5.1 ± 0.2 25.3 ± 0.3	- 100.8 ± 3.7	- 20	160 ± 4 178 ± 5	- 98.9 ± 2.8
LCD Phone 2	- 20	287 ± 2 306 ± 3	- 99.7 ± 3.7	- 20	412 ± 6 434 ± 6	- 100.4 ± 1.4
LCD Phone 3	- 20	190 ± 2 212 ± 4	- 100.9 ± 3.7	- 20	580 ± 6 596 ± 5	- 99.3 ± 0.9
Laptop screen 1	- 20	215 ± 3 236 ± 4	- 100.4 ± 3.7	- 20	360 ± 5 378 ± 5	- 99.5 ± 1.3
Laptop screen 2	- 20	410 ± 2 432 ± 4	- 100.4 ± 3.7	- 20	265 ± 4 283 ± 5	- 99.3 ± 1.7
	**Ga(III), µg g^−1^**			**In(III), µg g^−1^**		
**Leaching**	**Added**	**Found ^a^**	**Rec., %**	**Added**	**Found ^a^**	**Rec., %**
LED light bulb	- 20	730 ± 3 749 ± 4	- 99.8 ± 0.5	- 20	20 ± 2 41 ± 2	- 102.5 ± 4.9
LCD phone 1	- 20	3.8 ± 0,2 23.3 ± 0,3	- 97.9 ± 1.3	- 20	112 ± 3 133 ± 4	- 100.7 ± 3.0
LCD phone 2	- 20	280 ± 2 302 ± 3	- 100.7 ± 0.9	- 20	403 ± 6 424 ± 6	- 102.0 ± 1.4
LCD phone 3	- 20	184 ± 2 205 ± 4	- 100.5 ± 1.9	- 20	551 ± 5 569 ± 5	- 99.6 ± 0.9
Laptop screen 1	- 20	198 ± 4 223 ± 5	- 102.3 ± 2.2	- 20	327 ± 4 349 ± 5	- 100.6 ± 1.4
Laptop screen 2	- 20	389 ± 4 411 ± 4	- 100.5 ± 0.9	- 20	252 ± 4 270 ± 5	- 99.2 ± 1.8

^a^ mean value ± standard deviation of three determinations.

## Data Availability

Data are available from the corresponding authors upon reasonable request.

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
