# Peer review of "Using a Simple Magnetic Adsorbent for the Preconcentration and Determination of Ga(III) and In(III) by Electrothermal Atomic Absorption Spectrometry"

_molecules, 2023, doi:10.3390/molecules28062549_

Round 1
Reviewer 1 Report
In this work, the authors used Fe3O4 adsorbent for preconcentration and determination of Ga(III) and In(III), this work is systematically performed and the results are also satisfied, so I suggest acceptance after addressing these concerns.
1. Line 27-49, the background for Ga(III) and In(III) is too long and not the key point of this work, why Ga(III) and In(III) were extensively utilized and why they were discharged into wastewater/soil are of the most importance.
2. Line 50-58, the methods for Ga(III) and In(III) elimination should be systematically described.
3. I think the authors should clarify their innovation in the introduction, the adsorption by magnetic particles have been extensively reported, so if the new ETAAS integrated with DMSPE for Ga(III) and In(III) detection is the point, Table 2 also lists many ETAAS research with different adsorbents, I think it’s not sufficient.
4. Line 84-94, chitosan and oleic acid coated magnetic particles are mentioned in solid-phase extraction rather than DMSPE, however, these two materials are generally designed as highly dispersive nanoparticles and the authors should check the ref, and for the other materials for SPE, their mechanism might also be adsorption so the authors should revise these two paragraphs.
5. Line 312, coprecipitation may be more accurate.
6. The authors should detect the particle size distribution to examine the dispersion of the nanoparticles.
7. Line 104, what’s the surface state of Fe3O4 and exist states of Ga(III) and In(III) in this pH range?
8. Line 162, the authors use acid to desorb, if the Fe3O4 would be partially dissolved and if the coexisting Fe(III) would influence the results?
9. Table 2, the enrichment factor is 163 ± 2 while it’s 160 in this table?
Author Response
Response to reviewer 1
In this work, the authors used Fe3O4 adsorbent for preconcentration and determination of Ga(III) and In(III), this work is systematically performed and the results are also satisfied, so I suggest acceptance after addressing these concerns.
Thank you very much for your good comments on the text of the paper we have submitted. We are sure that they substantially improve its content. We have addressed all your comments. The response to each of them is detailed below
- Line 27-49, the background for Ga(III) and In(III) is too long and not the key point of this work, why Ga(III) and In(III) were extensively utilized and why they were discharged into wastewater/soil are of the most importance.
- Line 50-58, the methods for Ga(III) and In(III) elimination should be systematically described.
1 and 2. According to your indication we have completely modified the Introduction of the text and a new wording has been presented. We hope that your comments have been taken on board.
- I think the authors should clarify their innovation in the introduction, the adsorption by magnetic particles have been extensively reported, so if the new ETAAS integrated with DMSPE for Ga(III) and In(III) detection is the point, Table 2 also lists many ETAAS research with different adsorbents, I think it’s not sufficient.
We have gone into the details of the various procedures published for the pre-concentration of Ga and In. Please note that this has been done in such a way that the same published work is discussed at several points throughout the development of sections 1 and 2 of the text.
- Line 84-94, chitosan and oleic acid coated magnetic particles are mentioned in solid-phase extraction rather than DMSPE, however, these two materials are generally designed as highly dispersive nanoparticles and the authors should check the ref, and for the other materials for SPE, their mechanism might also be adsorption so the authors should revise these two paragraphs.
We agree with your comment and have corrected this error in the amended text.
- Line 312, coprecipitation may be more accurate.
We agree and it has been amended accordingly.
- The authors should detect the particle size distribution to examine the dispersion of the nanoparticles.
This comment, together with those posed by another reviewer, has prompted us to introduce a document containing supplementary material addressing your comment.
- Line 104, what’s the surface state of Fe3O4and exist states of Ga(III) and In(III) in this pH range?
We have inserted a small paragraph in the text directing the reader to the supplementary material.
- Line 162, the authors use acid to desorb, if the Fe3O4would be partially dissolved and if the coexisting Fe(III) would influence the results?
The partial solubilisation of the ferrite particles during the desorption process does not present any problems in the determination of In. In the case of Ga, as indicated in the text, the use of an alternative line (294.364 nm) to the resonance line (287.424 nm) is recommended, which does not entail a loss of sensitivity and ensures that the Fe signal does not overlap with that of Ga.
- Table 2, the enrichment factor is 163 ± 2 while it’s 160 in this table?
You are right. This error has been corrected.
Please note that we have enclosed the document with the revision marks.
Reviewer 2 Report
The manuscript presents a sound and careful study. The authors have considered all parameters that may affect extraction and validated their results with the analysis of many samples and certified materials. The procedures are described in detail (so that they can be reproduced by others) and the analytical results are good (detection limits, recoveries, etc).
Some minor issues should be addressed before acceptance:
1) Introduction. A statement explaining why its material was selected should be added. The authors report the use of various materials including magnetic nanoparticles (in the Introduction and Table 2). Why use pure magnetic nanoparticles? (they are cheap, easy to produce in mass quantities and in a reproducible manner, other reason ?).
2) Basic characterization studies of the magnetic nanoparticles are not reported. Although this a pure analytical work some information about the material should be included and discussed such as the XRD pattern, the zero point of charge, magnetization studies (VSM), particle size distribution (TEM or SEM), etc. Other data such as TFA/BET may be also included. It is important to know the basic properties of the nanomaterial so that the work can be reproduced by others.
3) What is the maximum sample volume that can be extracted with this method and material without impairing the analytical data? The use of larger sample volume (not only 10 mL) may significantly improve the LOD of the method. Why not try to extract a larger sample volume? In such samples (soil extracts, water, extracts from electronic devices), the sample volume is not a limitation. This information can be added in the discussion without changing the structure of the paper.
4) A few typographical errors should be corrected:
Table 5-Calcination-LED light bulb: The recovery of In(III) should be 102.5 (not 1025)
Table 5-Leaching- LCD phone 2: The recovery of In(III) should be 102 (not 10.,2)
Table 4-NIST SRM 2711b (soil): No SD value is given for the certified value of Ga(III).
Table 3 and Table 5. In Table 3 recoveries are given with the SD value, while in Table 5 only the recoveries are reported. Please use one format (with or without SD).
Author Response
Response to reviewer 2
The manuscript presents a sound and careful study. The authors have considered all parameters that may affect extraction and validated their results with the analysis of many samples and certified materials. The procedures are described in detail (so that they can be reproduced by others) and the analytical results are good (detection limits, recoveries, etc).
Thank you for your good comments on the text of our submitted paper. We are sure that they improve its content considerably. We have considered all your comments. The response to each of them is detailed below
Some minor issues should be addressed before acceptance:
1) Introduction. A statement explaining why its material was selected should be added. The authors report the use of various materials including magnetic nanoparticles (in the Introduction and Table 2). Why use pure magnetic nanoparticles? (they are cheap, easy to produce in mass quantities and in a reproducible manner, other reason ?).
In accordance with your appreciation, we have completely modified the introduction of the text and a new wording has been presented. We hope that your comments have been taken into account.
2) Basic characterization studies of the magnetic nanoparticles are not reported. Although this a pure analytical work some information about the material should be included and discussed such as the XRD pattern, the zero point of charge, magnetization studies (VSM), particle size distribution (TEM or SEM), etc. Other data such as TFA/BET may be also included. It is important to know the basic properties of the nanomaterial so that the work can be reproduced by others.
This comment, together with a comment made by another reviewer, has led us to introduce a document with additional material that includes your comment. We have introduced new paragraphs in sections 2.1 and 4.2 to elaborate on these details and to direct the reader to the supplementary material.
3) What is the maximum sample volume that can be extracted with this method and material without impairing the analytical data? The use of larger sample volume (not only 10 mL) may significantly improve the LOD of the method. Why not try to extract a larger sample volume? In such samples (soil extracts, water, extracts from electronic devices), the sample volume is not a limitation. This information can be added in the discussion without changing the structure of the paper.
The amount of adsorbent indicated in the text has been optimised for 10 mL of sample. However, it is possible to work with larger volumes. We have verified that the retention remains quantitative in less than 10 min contact time when 25 mL of sample were treated with 200 µl of ferrite suspension. Larger volumes are of little interest from an analytical point of view because they require a change in the volume of the desorption medium of the retained analytes. Otherwise the desorption is not quantitative and the increase in the enrichment factor is lost.
4) A few typographical errors should be corrected:
Table 5-Calcination-LED light bulb: The recovery of In(III) should be 102.5 (not 1025)
Table 5-Leaching- LCD phone 2: The recovery of In(III) should be 102 (not 10.,2)
Table 4-NIST SRM 2711b (soil): No SD value is given for the certified value of Ga(III).
Table 3 and Table 5. In Table 3 recoveries are given with the SD value, while in Table 5 only the recoveries are reported. Please use one format (with or without SD).
Thank you for your observations. All the above errors have been corrected. We have revised the text in more detail and made some more corrections.
Please note that we have enclosed the document with the revision marks
Round 2
Reviewer 1 Report
This work has been well revised and I suggest acceptance now.